# Learning to Cool:
# State Space Models for Smarter Automobile AC Systems

**Sam Mikhak** [1]   **Forrest Felsch** [2]   **Taehyung Wang** [3]

## Abstract

We investigate the application of modern state-space models for time series forecasting of automobile air conditioning (AC) power draw, a task critical for improving heating, ventilation, and air conditioning (HVAC) control systems impacting fuel or battery consumption by up to 20% (Vasile-Müller, 2011). Traditional long short-term memory (LSTM) based approaches struggle with long-range dependencies in high-resolution vehicle telemetry, motivating our exploration of the Structured State Space Sequence (S4) architecture. We train and evaluate S4 forecaster variants on more than 7,000 sliding windows of data, then compare them against a hybrid LSTM-Attention baseline. Across multiple window configurations, S4 consistently outperforms the baseline, achieving a mean squared error (MSE) as low as 0.000288, an $R^2$ up to 0.9820, and a weighted mean absolute percentage error (WMAPE) as low as 0.0247. We also report latency and parameter counts, showing that the S4 forecaster remains compact and suitable for in-vehicle deployment. These results demonstrate that structured state-space models can improve predictive accuracy for automotive HVAC forecasting while maintaining a lightweight footprint, providing a foundation for next-generation energy-aware control systems.

## 1. Introduction

Air conditioning (AC) is a major contributor to auxiliary energy use in both internal combustion engine (ICE) vehicles and electric vehicles (EVs). In ICE cars, running the AC can increase fuel consumption by up to 20% while in EVs it can reduce driving range by 10–15% (Vasile-Müller, 2011). Accurate, real-time forecasting of AC power draw is therefore essential for intelligent HVAC control and overall energy optimization.

Most prior work has relied on recurrent neural networks such as long short-term memory (LSTM) models, sometimes augmented with Attention mechanisms, to predict AC load from vehicle telemetry (Tazhibi et al., 2024; Xu et al., 2021). Although these models achieve reasonable short-term accuracy, they often struggle to capture very long-range dependencies when sampling at high frequency over extended drives.

In this paper, we explore structured state-space sequence (S4) models as an alternative for AC power forecasting. S4 architectures offer a principled way to model continuous-time dynamics, long-range dependencies, and efficient discretization in a single unified framework (Gu et al., 2022a). We adapt the S4 forecaster to our vehicle-level AC dataset and compare it against hybrid LSTM-Attention baselines across multiple forecasting horizons.

## 2. Related Work

Several recent studies have applied machine learning to understand and predict the energy consumption patterns of automotive air conditioning (AC) systems.

Tazhibi et al. (2024) explored the predictive modeling of power consumption in automotive AC systems using XG-Boost and ensemble regressors, highlighting the significance of compressor dynamics and thermal conditions. Their work demonstrates that supervised ML pipelines can achieve high accuracy (e.g., RMSE $< 0.1\%$) in forecasting fuel usage during AC operation, especially when enriched with rolling statistics and external weather data. However, their models were largely based on tree ensembles and did not leverage long-range temporal dependencies across time-series inputs.

Kaddoura et al. (2024) extended this direction by incorporating a stacking ensemble of decision tree-based regressors and classifiers to analyze the fuel and emissions impact of AC usage across hybrid and internal combustion engine vehicles. Using 37 OBD-II features refined to 29 through

[1]Columbia University, New York, NY, USA [2]Oregon State University, Corvallis, OR, USA [3]Department of Computer Science, California State University, Northridge, CA, USA. Correspondence to: Taehyung Wang <twang@csun.edu>.

*Proceedings of the $2^{nd}$ ICML Workshop on Foundation Models for Structured Data*, Seoul, South Korea. 2026. Copyright 2026 by the author(s).

feature selection, they reported 99% classification accuracy and showed the importance of $CO_2$ and NOx emissions as predictive features. Notably, they used static and dynamic features alongside weather data collected via OpenWeatherMap, but their approach did not address the modeling of sequential dependencies explicitly.

Kaddoura et al. (2025) focused on high-resolution OBDLink-MX+ data to predict power draw from AC usage in a Toyota Corolla LE, using tree-based regressors trained on feature-engineered signals. Their findings established that temperature, compressor speed, and engine load are strong indicators of AC load. While informative, the models lacked the expressive power of structured sequence models and were sensitive to feature redundancy and window size.

In contrast to these works, our approach uses a state-of-the-art sequence modeling architecture, the Structured State-Space (S4) model, to directly learn long-range dependencies in high-frequency vehicle telemetry. By combining HiPPO-derived Legendre-state dynamics with convolution-like updates, our method offers a dynamics-motivated inductive bias while achieving low inference latency and strong accuracy. Our work extends prior research by evaluating S4-based forecasters on various horizons, demonstrating robust performance under multi-step settings.

# 3. Methodology

### 3.1. Data Preparation

We use an OBDLink-MX+ telemetry dataset collected on a hybrid 2013 Toyota Prius V. After synchronizing CAN-bus channels and resampling to 1Hz, we select 30 telemetry features (engine speed, intake temperature, vehicle speed, AC gate status, etc.) and compute rolling statistics over 5-step and 10-step windows. To expose short-term dynamics, we augment the feature vector with first-order temporal differences of all channels, concatenated to the original telemetry. Power draw is standardized via z-scoring. We split the first 80% of the timeline for training/validation and the remaining 20% for testing, avoiding overlap across sliding windows. All feature scaling and target standardization are fit on the training split only and applied to validation/test to avoid leakage.

Input-output pairs are generated using sliding windows of length `SEQ`, stride `STRIDE`, and forecast horizon `HOR` (non-overlapping when `STRIDE = SEQ`):

$$X_i = [u_i, u_{i+1}, \ldots, u_{i+\texttt{SEQ}-1}] \in \mathbb{R}^{\texttt{SEQ} \times d},$$

$$y_i = p_{i+\texttt{SEQ}+\texttt{HOR}-1},$$

where $u_t \in \mathbb{R}^d$ denotes the $d$-dimensional vehicle telemetry vector at time index $t$, and $p_t$ denotes the instantaneous AC power draw (kW) at time index $t$. Thus, the scalar target $y_i$ is the AC power draw (kW) at the future time index

$i+\texttt{SEQ}+\texttt{HOR}-1$. At 1Hz sampling, a horizon of `HOR` $= h$ corresponds to forecasting $h$ seconds into the future.

### 3.2. S4 Forecaster Architecture

Intuitively, each HiPPO-LegS block maintains a fixed-size memory that summarizes the entire input window at multiple temporal resolutions. Lower-order components of the state capture slow-varying trends such as cabin thermal inertia, while higher-order components respond to rapid fluctuations caused by compressor cycling or control changes. Stacking multiple LegS blocks allows the model to iteratively refine this multi-scale summary before producing a single-step power forecast.

Formally, we implement this process using a stacked S4 forecaster. The model projects each $X_i$ into an embedding, passes it through $L$ HiPPO-LegS blocks, and maps the output to a scalar prediction of AC electrical power draw (kW):

$$
\begin{aligned}
H_0 &= X_i W_{\text{in}} + b_{\text{in}}, \\
H_\ell &= \text{HiPPOLegSBlock}(H_{\ell-1}, X_i), \\
H_L &\to \text{Linear Head}.
\end{aligned}
$$

We sweep $L \in \{4, 8\}$, hidden size 512, and regularization $\lambda \in \{10^{-5}, 10^{-4}\}$.

The S4 backbone contains **49,770 trainable parameters**, and the regression head adds **2,305 parameters**, for a total of **52,075 parameters**.

### 3.3. Training and Evaluation Metrics

For this model, we use a weighted Huber loss

$$
\begin{aligned}
\mathcal{L}(y, \hat{y}) &= \frac{1}{N} \sum_{i=1}^{N} w_i \cdot \text{Huber}(y_i, \hat{y}_i; \delta), \\
w_i &= 1 + \min(C_{\text{max}}, |y_i| - \tau)_+^\gamma
\end{aligned}
$$

where sample weights are proportional to the magnitude of the target value, with Adam optimizer, cosine-decay learning rate, batch size 64, and gradient clipping (norm 1.0). Early stopping is based on validation loss. We found these hyperparameters to have the best training performance after a linear sampling of the space.

At test time, we evaluate on AC-on subsets of the data to mimic real deployment. We then also record latency and memory usage for deployment feasibility. To validate performance, we evaluate using three metrics: Mean Squared Error (MSE), Coefficient of Determination ($R^2$), and Weighted Mean Absolute Percentage Error (WMAPE). We use the first two metrics to keep consistent metrics with previous

works such as Kaddoura et al. (2025). We use the third metric to represent our error as a percentage while avoiding loss of precision error from near or at zero values. See Appendix C for formal definitions and further information on our metrics.

### 3.4. Hybrid LSTM-Attention Baseline

For a comparative baseline, we use a hybrid LSTM Multi-Headed Attention model proposed by Abbasimehr and Paki (Abbasimehr & Paki, 2021) and implemented via Py-Torch (Paszke et al., 2019). They found the hybrid model to perform better than LSTM, Multi-Headed Attention, and several other traditional time series models on 16 tested time series datasets. Please see (Abbasimehr & Paki, 2021) for further technical detail.

For our modeling, we use two different sizes. For direct comparison with the S4 model, we use a model with a similar number of trainable parameters, **50,920**. This model has 2 hidden layers of size 40 in the LSTM and 2 attention heads.

To attempt to maximize performance, we also use a model with **7,519,656** trainable parameters. This model has 4 hidden layers of size 512 in the LSTM and 10 attention heads.

## 4. Results

### 4.1. SSM S4 Forecasting Results

*Table 1.* SSM S4 Performance (mean $\pm$ std over 5 seeds) with 52,075 parameters

| SEQ | Stride | Hor | MSE | $R^2$ | WMAPE | Avg latency (s) |
|-----|--------|-----|-----|-------|-------|-----------------|
| 64 | 1 | 1 | **0.000288 $\pm$ 0.000067** | **0.9820 $\pm$ 0.0042** | **0.0247 $\pm$ 0.0016** | ~0.00018 |
| 32 | 1 | 1 | 0.000296 $\pm$ 0.000040 | 0.9707 $\pm$ 0.0025 | 0.0270 $\pm$ 0.0013 | ~0.00015 |
| 10 | 10 | 1 | 0.000353 $\pm$ 0.000085 | 0.9778 $\pm$ 0.0053 | 0.0252 $\pm$ 0.0016 | ~0.00015 |
| 5 | 5 | 1 | 0.000535 $\pm$ 0.000091 | 0.9665 $\pm$ 0.0057 | 0.0276 $\pm$ 0.0009 | ~0.00017 |
| 5 | 5 | 5 | 0.001146 $\pm$ 0.000021 | **0.9286 $\pm$ 0.0013** | **0.0581 $\pm$ 0.0015** | ~0.00029 |
| 10 | 10 | 5 | 0.001690 $\pm$ 0.000022 | 0.9054 $\pm$ 0.0014 | 0.0671 $\pm$ 0.0013 | ~0.00040 |
| 10 | 10 | 10 | **0.001679 $\pm$ 0.000013** | **0.8948 $\pm$ 0.0008** | **0.0764 $\pm$ 0.0003** | ~0.00049 |
| 5 | 5 | 10 | 0.001855 $\pm$ 0.000015 | 0.8845 $\pm$ 0.0009 | 0.0798 $\pm$ 0.0009 | ~0.00047 |
| 10 | 10 | 20 | **0.003015 $\pm$ 0.000044** | **0.8112 $\pm$ 0.0028** | **0.1129 $\pm$ 0.0014** | ~0.00052 |

The S4 model consistently achieves strong performance across all configurations. The best results are observed with the (64,1,1) and (32,1,1) settings, which achieve MSEs of **0.000288** and **0.000296**, with corresponding $R^2$ values of **0.9820** and **0.9707**. These configurations also yield the lowest WMAPE values, indicating that the model maintains small relative errors in addition to low absolute prediction error. Together, these results highlight the models ability to capture long-range temporal dynamics effectively when dense sampling is available.

Furthermore, the model also maintains robust performance under reduced temporal redundancy, as seen in the (10,10,1) and (5,5,1) configurations, which re-

tain high predictive accuracy with $R^2$ values above 0.96 and similarly low WMAPE values. For moderate forecast horizons of 5 steps, performance degrades gracefully: (5,5,5) achieves an $R^2$ of **0.9286** with a WMAPE of **0.0581**, while the sparser (10,10,5) configuration attains $R^2 = 0.9054$.

For longer horizons, accuracy naturally declines but remains stable. At a 10-step horizon, (10,10,10) achieves $R^2 = $ **0.8948** with a WMAPE of **0.0764**, while (5,5,10) yields a comparable $R^2 = 0.8845$. Even at a 20-step horizon, (10,10,20), the model preserves meaningful predictive power with an $R^2 = $ **0.8112** and WMAPE of **0.1129**, demonstrating that the S4 forecaster generalizes across a wide range of temporal contexts and multiple forecasting horizons. A graph of the forecasting can be found in Appendix E.

### 4.2. Hybrid LSTM-Attention Baseline

*Table 2.* Hybrid Performance (mean $\pm$ std over 5 seeds) with 50,920 parameters

| SEQ | Stride | Hor | MSE | $R^2$ | WMAPE | Avg latency (s) |
|-----|--------|-----|-----|-------|-------|-----------------|
| 64 | 1 | 1 | **0.004418 $\pm$ 0.000984** | **0.6345 $\pm$ 0.0814** | **0.3200 $\pm$ 0.0479** | ~0.00035 |
| 32 | 1 | 1 | 0.002696 $\pm$ 0.000881 | 0.7768 $\pm$ 0.0729 | 0.2240 $\pm$ 0.0277 | ~0.00032 |
| 10 | 10 | 1 | 0.001512 $\pm$ 0.000629 | 0.8742 $\pm$ 0.0523 | 0.1878 $\pm$ 0.0135 | ~0.00033 |
| 5 | 5 | 1 | **0.000915 $\pm$ 0.000118** | **0.9241 $\pm$ 0.0098** | **0.1473 $\pm$ 0.0334** | ~0.00032 |
| 5 | 5 | 5 | 0.002145 $\pm$ 0.000719 | 0.8230 $\pm$ 0.0593 | 0.2250 $\pm$ 0.0243 | ~0.00032 |
| 10 | 10 | 5 | 0.002670 $\pm$ 0.001548 | 0.7805 $\pm$ 0.1272 | 0.2108 $\pm$ 0.0304 | ~0.00032 |
| 10 | 10 | 10 | 0.002887 $\pm$ 0.000368 | 0.7609 $\pm$ 0.0305 | 0.2352 $\pm$ 0.0081 | ~0.00033 |
| 5 | 5 | 10 | 0.002393 $\pm$ 0.000452 | 0.8026 $\pm$ 0.0373 | 0.2484 $\pm$ 0.0212 | ~0.00031 |
| 10 | 10 | 20 | 0.003826 $\pm$ 0.001389 | 0.6832 $\pm$ 0.1151 | 0.2584 $\pm$ 0.0202 | ~0.00032 |

For a direct comparison, we test a Hybrid model with a similar parameter count to the S4 model. In the best case (5,5,1), as seen in Table 2, it achieves an MSE of **0.000915**, $R^2$ of **0.9241**, and WMAPE of **0.1473**. However, most other configurations trail significantly further behind. In the worst case (64,1,1), it achieves an MSE of **0.004418**, $R^2$ of **0.6345**, and WMAPE of **0.3200**. In comparison, in the same configuration the S4 achieves an MSE of **0.000288**, $R^2$ of **0.9820**, and WMAPE of **0.0247**. This supports the advantages of SSMs for modeling long-term and multiscale temporal dependencies inherent in automotive HVAC dynamics.

In an attempt to maximize performance, we also evaluate a hybrid model with 7,519,656 parameters. Yet this model also performs worse across all predictive metrics and configurations. In the best case (5,5,1), as seen in Appendix D Table 3, it achieves an MSE of **0.000562**, $R^2$ of **0.9534**, and WMAPE of **0.0924**. All of which trail both behind the S4s corresponding values of 0.000535, 0.9665, and 0.0276 respectively despite using **144 times** the parameters.

## 5. Limitations

This work should be regarded as a proof of concept for applying structured state-space models to automotive AC power consumption forecasting. While our approach demonstrates strong performance and improved efficiency in modeling sequential dependencies, several factors limit its current scope. First, the experiments were conducted on data from a specific set of vehicles and driving conditions, which may not fully capture the diversity of real-world environments. Second, our evaluation primarily focused on short-to medium-term forecasting horizons; extending the framework to significantly longer horizons represents the next phase of our research. Lastly, real-world deployment would require additional robustness measures, including strategies for handling sensor noise, missing data, and domain shifts over time. These considerations present opportunities for future research to extend the applicability of this framework.

## 6. Conclusion and Future Work

In this paper, we demonstrate the effectiveness of Structured State-Space Sequence (S4) models for forecasting automobile air conditioning (AC) power consumption from high-resolution vehicle telemetry. Our results show that the S4 forecaster consistently outperforms hybrid LSTM-Attention baselines across multiple windowing configurations and evaluation regimes. Achieving an MSE of as low as $0.000288$, an $R^2$ up to $0.9820$, and a WMAPE as low as $0.0247$, the model effectively captures both global trends and localized variations in AC load. Furthermore, its computational efficiency and compact memory footprint make it a strong candidate for real-time in-vehicle deployment.

For future work, we plan to extend our approach in several directions. First, we aim to explore adaptive and nonlinear variants of SSMs such as Liquid-S4 or Mamba to better handle context-dependent AC behavior. Second, we intend to incorporate external environmental factors (e.g., ambient temperature, humidity, solar radiation) to enable more anticipatory control strategies. Third, integrating forecasting outputs into closed-loop HVAC controllers will allow us to evaluate energy savings and user comfort in real-world driving scenarios. Fourth, we aim to validate our approach on data from fully electric vehicles such as the Tesla Model 3 or Model Y, which exhibit different thermal and energy management characteristics compared to hybrid platforms. Finally, we are interested in multi-task extensions that jointly model compressor control, cabin temperature, and battery thermal load for holistic energy management in electric vehicles.

## Acknowledgments

This work was conducted as part of the CSUN REU program on Edge Computing and Data Science. We thank our mentors Dr. Taehyung Wang, Dr. Xunfei Jiang, Kiranmayee Lokam, and Hemanth Kumar Tulabandula from California State University, Northridge for their time and guidance throughout the project. This research is supported by the National Science Foundation under Grant CNS-2244391

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

# A. Appendix

Additional experiments, model configurations, and ablations can be found here.

# B. Background

### B.1. Structured State-Space Sequence (S4) Models

Structured State-Space Sequence (S4) models marry continuous-time state-space modeling with efficient discrete implementation. Given an input sequence $u_{1:T}$, represented in continuous time as $u(t)$, an S4 layer views it as driving a linear time-invariant system:

$$\frac{d}{dt}h(t) = Ah(t) + Bu(t), \quad y(t) = Ch(t) + Du(t),$$

where $h(t) \in \mathbb{R}^N$ is a hidden memory state and $A, B, C, D$ are learned. By choosing the transition matrix $A$ according to the HiPPO principle and applying a low-rank Legendre-kernel discretization, S4 achieves:

- **Principled dynamics:** The continuous ODE summarizes all past inputs without ad-hoc truncation.

- **Long-range retention:** HiPPO-derived $A$ is optimally tuned to retain long-range information more effectively than standard recurrent updates.

- **Fast single-layer updates:** The specialized discretization collapses the recurrence into a convolution-like layer.

### B.2. HiPPO Legendre-State (LegS) Blocks

The core of S4 is the HiPPO-LegS block, which implements a discrete recurrence:

$$h_{k+1} = \bar{A}h_k + \bar{B}u_k, \quad y_k = \bar{C}h_k + \bar{D}u_k,$$

where the matrices $\bar{A}, \bar{B}, \bar{C}, \bar{D}$ are parameterized to encode projections of the input history onto an orthonormal Legendre basis. Each block compresses the entire input history. Stacking $L$ such blocks yields an S4 layer that can capture fast spikes and slow drifts in time series like AC power draw.

### B.3. Computational and Memory Efficiency of S4

As established in prior work Gu et al. (2022b;a), the Structured State Space (S4) model offers several efficiency advantages over recurrent architectures such as LSTM and hybrid LSTM-Attention models. These properties are a result of the continuous-time state-space formulation and do not depend on our specific implementation or dataset. In particular, S4 provides:

- **Fixed-size memory.** The recurrent state in S4 remains constant in size regardless of sequence length. This avoids the memory overhead that LSTMs encounter when unrolled over thousands of timesteps.

- **Efficient convolutional representation.** Through a mathematical transformation, the recurrent updates in S4 can be expressed as a single convolution with a learned, fixed kernel. This allows computations to be parallelized across timesteps and reduces runtime complexity during both training and inference.

- **Scalability to long sequences.** Because the convolutional kernel can be precomputed, S4 scales efficiently to long horizons without the need for truncated backpropagation through time. This enables lower computational cost while maintaining high predictive performance.

Together, these properties enable S4 not only to outperform LSTM and hybrid LSTM-Attention baselines in predictive accuracy, but also to achieve greater computational and memory efficiency. This makes S4 especially well-suited for real-world deployment in long sequence forecasting tasks.

## C. Metrics

For the data points in our testing set, we denote the true value $y_i$, the predicted value $\hat{y}_i$, and the number of data points $N$. We use the first two metrics to keep consistent metrics with previous works such as Kaddoura et al. (2025). We use Mean Squared Error with the definition

$$MSE := \frac{1}{N} \sum_{i=1}^{N} (y_i - \hat{y}_i)^2$$

in order to evaluate the average error of our model. We use Coefficient of Determination with the definition

$$R^2 := 1 - \frac{\sum_{i=1}^{N} (y_i - \hat{y}_i)^2}{\sum_{i=1}^{N} (y_i - \frac{1}{N} \sum_{i=1}^{N} y_i)^2}$$

in order to evaluate how well our sequence of predicted values fits the sequence of actual values. Finally, we use Weighted Mean Absolute Percentage Error with the definition

$$WMAPE := \frac{\sum_{i=1}^{N} |y_i - \hat{y}_i|}{\sum_{i=1}^{N} |y_i|}$$

in order to evaluate our model's error as a percentage while avoiding loss of precision errors caused by near zero values. Together these three metrics provide a comprehensive view into the performance of our model.

## D. Large Hybrid Model Performance

*Table 3.* Hybrid Performance (mean ± std over 5 seeds) with 7,519,656 parameters.

| SEQ | Stride | Hor | MSE | $R^2$ | WMAPE | Avg latency (s) |
|---|---|---|---|---|---|---|
| 64 | 1 | 1 | $0.002765 \pm 0.000366$ | $0.7713 \pm 0.0303$ | $0.3024 \pm 0.0456$ | $\sim 0.00293$ |
| 32 | 1 | 1 | $0.001894 \pm 0.000305$ | $0.8432 \pm 0.0252$ | $0.1926 \pm 0.0294$ | $\sim 0.00171$ |
| 10 | 10 | 1 | $0.000728 \pm 0.000256$ | $0.9394 \pm 0.0213$ | $0.1266 \pm 0.0090$ | $\sim 0.00107$ |
| 5 | 5 | 1 | $\mathbf{0.000562 \pm 0.000097}$ | $\mathbf{0.9534 \pm 0.0080}$ | $\mathbf{0.0924 \pm 0.0132}$ | $\mathbf{\sim 0.00098}$ |
| 5 | 5 | 5 | $0.001398 \pm 0.000270$ | $0.8847 \pm 0.0223$ | $0.1280 \pm 0.0075$ | $\sim 0.00082$ |
| 10 | 10 | 5 | $0.001474 \pm 0.000344$ | $0.8788 \pm 0.0283$ | $0.1636 \pm 0.0237$ | $\sim 0.00102$ |
| 10 | 10 | 10 | $0.002067 \pm 0.000349$ | $0.8288 \pm 0.0289$ | $0.1825 \pm 0.0124$ | $\sim 0.00102$ |
| 5 | 5 | 10 | $0.001996 \pm 0.000383$ | $0.8353 \pm 0.0316$ | $0.1737 \pm 0.0197$ | $\sim 0.00082$ |
| 10 | 10 | 20 | $0.002513 \pm 0.000476$ | $0.7919 \pm 0.0395$ | $0.1979 \pm 0.0110$ | $\sim 0.00102$ |

Table of the large hybrid model. Results are discussed in Section 4.2.

The following section contains graphs illustrating model forecasting of AC power draw compared to actual AC power draw.

The proposed hybrid model first computes the LSTM representation and the Multi-Headed Attention representation. Then it concatenates these representations with the original input data and predicts the output value, AC electrical power draw (kW), using a dense fully connected layer. We implement this using the PyTorch Library's built-in LSTM and MultiheadAttention classes (Paszke et al., 2019). In order to make a consistent comparison, this model uses the same training loss and hyperparameters as the S4 model, which are defined in section 3.3.

## E. Forecasting Graphs

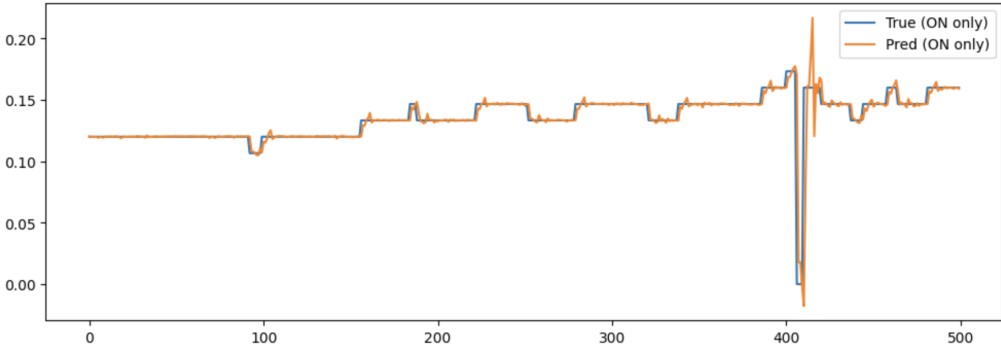

*Figure 1.* S4 Model with SEQ = 32, Stride = 1, and Horizon = 1.

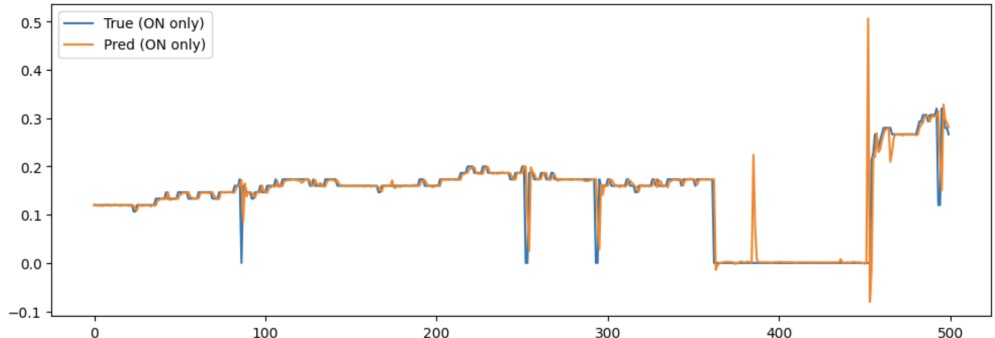

*Figure 2.* S4 Model with SEQ = 5, Stride = 5, and Horizon = 5.

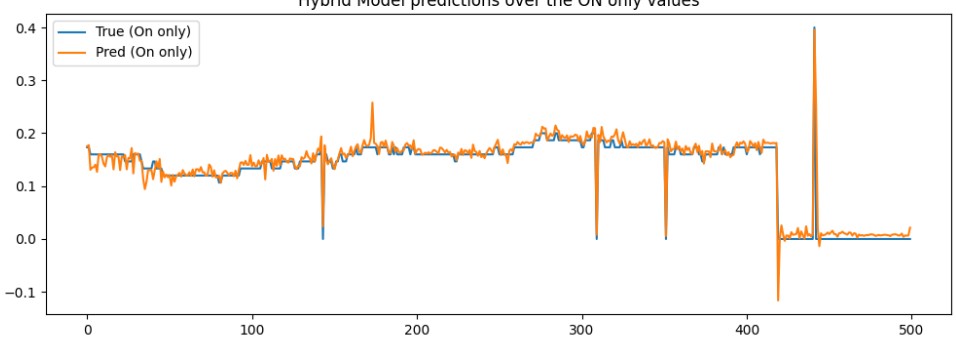

*Figure 3.* Hybrid model with 7,519,656 parameters and SEQ = 5, Stride = 5, and Horizon = 1.

