# OpenReview forum: "Learning to Cool: State Space Models for Smarter Automobile AC Systems"
_ICML.cc/2026/Workshop/FMSD — FMSD @ ICML 2026 Poster_

### Official Review · Reviewer_pzYB · 2026-05-19
**Good case study work showing the superiority of S4 models for automobile AC power forecasting**

**Rating:** 6
**Confidence:** 4

**Review:**

This work demonstrates the effectiveness of Structured State-Space Sequence (S4) models for forecasting automobile air-conditioning (AC) power consumption using high-resolution vehicle telemetry data. The results show that the S4 forecaster outperforms hybrid LSTM-Attention baselines across multiple windowing configurations and evaluation settings. Overall, the paper is easy to follow and presents strong experimental results. However, the reviewer has several concerns:

1)The writing quality should be improved, as the manuscript contains a number of grammatical and language issues.
2)The case studies are conducted using data collected from a single device. The reviewer encourages the authors to evaluate the method on multiple devices or larger-scale datasets to better demonstrate the generalizability of the approach.
3)For the ablation studies involving different settings (e.g., sequence length, stride, and forecasting horizon), the authors should fix one variable at a time and analyze the resulting performance variations. Further discussion of the potential reasons behind these trends would strengthen the impact of the case studies.

---

### Official Review · Reviewer_VdNX · 2026-05-20
**Learning to Cool: State Space Models for Smarter Automobile AC Systems**

**Rating:** 4
**Confidence:** 4

**Review:**

The authors present an application of Structured State Space Sequence models to forecast automobile air conditioning power consumption using high resolution vehicle telemetry. Addressed the computational and performance limitations of traditional recurrent neural networks (like LSTMs) on long range, high frequency dependencies, the study compares an S4 forecaster utilizing HiPPO-LegS blocks against a hybrid LSTM-Attention baseline. Using OBD-II data collected from a 2013 Toyota Prius V, evaluating the models across many sequence lengths, strides, and forecast horizons. The results show that a compact 52k-parameter S4 model achieves superior predictive accuracy (measured by MSE, WMAPE) compared to a LSTM Attention baseline.

Strengths
1. relevant application of S4 models to HVAC optimization for energy savings.
2. good evaluation matrix across multiple temporal horizons and window configurations.

Areas for Improvement

1. Relying on only one dataset 2013 Toyota Prius V limits generalizability, especially for full EVs.
2. The hybrid LSTM baseline is slightly dated; comparisons to modern time-series Transformers or linear baselines could be better

---

### Official Review · Reviewer_4CEK · 2026-05-22
**Review for Submission 157**

**Rating:** 7
**Confidence:** 4

**Review:**

Review:

The paper investigates the use of Structured State Space Sequence Models (S4) for forecasting automobile air conditioning power draw from high-resolution vehicle telemetry. The authors motivate the problem well: AC usage can significantly affect fuel or battery consumption, making accurate forecasting important for intelligent HVAC control and energy optimization. The paper compares S4 forecasters against hybrid LSTM-Attention baselines across multiple sequence lengths, strides, and forecasting horizons. The results show that S4 achieves strong predictive performance, with the best configuration reaching MSE \(=0.000288\), \(R^2=0.9820\), and WMAPE \(=0.0247\), while maintaining a compact model size of around 52K parameters and low inference latency.

Strengths:

(1) The paper addresses a practically important problem. Forecasting automobile AC power consumption is relevant for both ICE vehicles and EVs, where HVAC can meaningfully affect energy efficiency and driving range.

(2) The use of S4 models is well motivated. The paper clearly explains why traditional LSTM-based approaches may struggle with long-range dependencies in high-frequency telemetry data, and why structured state-space models are a promising alternative.

(3) The methodology is reasonably detailed. The paper describes data preparation, feature construction, sliding-window generation, train/test split, S4 architecture, training loss, and evaluation metrics.

(4) The empirical results are strong. S4 consistently outperforms both a parameter-matched hybrid LSTM-Attention baseline and a much larger hybrid baseline. In particular, the large hybrid model has about 7.5M parameters, yet still trails the 52K-parameter S4 model in accuracy and WMAPE.

(5) The paper considers deployment feasibility by reporting parameter count and inference latency, which is important for real-time in-vehicle forecasting.

Areas for Improvement:

(1) The dataset scope is narrow. The experiments appear to be based on telemetry from a hybrid 2013 Toyota Prius V, so it is unclear how well the method generalizes across vehicle types, climates, driving behaviors, and HVAC systems.

(2) The baselines could be broader. The paper compares against hybrid LSTM-Attention models, but does not include classical baselines such as XGBoost or random forests, or recent sequence models such as TCN, Transformer, Informer, or Mamba.

(3) The paper would benefit from ablation studies. It is not fully clear which components contribute most to the gains: S4 architecture, rolling statistics, first-order differences, weighted Huber loss, or AC-on-only evaluation.

(4) The evaluation focuses mainly on prediction accuracy, but the connection to downstream HVAC control is still indirect. It would be valuable to show whether the forecasting improvements translate into reduced energy consumption, better comfort, or more stable control behavior.

Detailed Comments:

(1) Please clarify the dataset size and collection setting more explicitly, including total drive duration, number of trips, environmental variation, and train/test split statistics.

(2) Please clarify whether training is performed on AC-on-only data or on both AC-on and AC-off data. This matters because AC-off periods may change the target distribution and make the forecasting task easier.

(3) Please add ablation studies for the feature engineering steps, especially rolling statistics and first-order temporal differences, to better isolate the benefit of S4 itself.

(4) The weighted Huber loss is reasonable, but the values of \(\tau\), \(\gamma\), and \(C_{\max}\) should be reported for reproducibility.

(5) Please discuss why the parameter-matched hybrid LSTM-Attention baseline performs especially poorly in dense sequence settings such as \(\text{SEQ}=64\), \(\text{stride}=1\), and \(\text{horizon}=1\). This result is important but somewhat surprising.

(6) It would be useful to include confidence intervals or statistical tests comparing S4 and the baselines, beyond mean and standard deviation over 5 seeds.

Justification of Score:
The paper addresses an important applied forecasting problem and presents clear evidence that S4 models can outperform LSTM-Attention baselines while being compact and fast enough for possible in-vehicle use. However, the paper is still closer to a strong proof-of-concept than a fully mature study because the dataset scope is limited, the baseline set is narrow, and ablations are missing.